# Clay-Based Hydrogels as Drug Delivery Vehicles of Curcumin Nanocrystals for Topical Application

**DOI:** 10.3390/pharmaceutics14122836

**Published:** 2022-12-17

**Authors:** Marco Ruggeri, Rita Sánchez-Espejo, Luca Casula, Raquel de Melo Barbosa, Giuseppina Sandri, Maria Cristina Cardia, Francesco Lai, César Viseras

**Affiliations:** 1Department of Drug Sciences, University of Pavia, Viale Taramelli 12, 27100 Pavia, Italy; 2Department of Pharmacy and Pharmaceutical Technology, University of Granada, 18071 Granada, Spain; 3Unit of Drug Sciences, Department of Life and Environmental Sciences, University of Cagliari, 09124 Cagliari, Italy

**Keywords:** nanocrystals, curcumin, clays, hydrogel, skin permeation

## Abstract

The poor water solubility of a significant number of active pharmaceutical ingredients (API) remains one of the main challenges in the drug development process, causing low bioavailability and therapeutic failure of drug candidates. Curcumin is a well-known Biopharmaceutics Classification System (BCS) class IV drug, characterized by lipophilicity and low permeability, which hampers topical bioavailability. Given these premises, the aim of this work was the design and the development of curcumin nanocrystals and their incorporation into natural inorganic hydrogels for topical application. Curcumin nanocrystals were manufactured by the wet ball milling technique and then loaded in clay-based hydrogels. Bentonite and/or palygorskite were selected as the inorganic gelling agents. Curcumin nanocrystal-loaded hydrogels were manufactured by means of a homogenization process and characterized with respect to their chemico-physical properties, in vitro release, antioxidant activity and skin permeation. The results highlighted that the presence of bentonite provided an increase of curcumin skin penetration and simultaneously allowed its radical scavenging properties, due to the desirable rheological characteristics, which should guarantee the necessary contact time of the gel with the skin.

## 1. Introduction

Recent advances in combinatorial organic synthesis, high-throughput screening, biotechnology and genomics have led to higher productivity in pharmaceutical research and development, creating a substantial increase in the number of molecules which could be considered as potential drug candidates [1]. However, many of these Active Pharmaceutical Ingredients fall under class II and IV of the BCS, representing approximately 70% of drugs and active entities [2,3]. In fact, the low water solubility of a significant number of APIs remains one of the main challenges in the drug development process, causing low bioavailability and therapeutic failure of drug candidates.

Conventional approaches to increasing the dissolution of poorly water-soluble compounds include salt formation, use of solubilizing agents, and addition of co-solvents [4]. However, the International Council for Harmonisation (ICH) Q3C guidelines on impurities (guidelines for residual solvents) limits the use of solvents in pharmaceutical applications.

With the aim of overcoming these issues, several attempts have been made to improve solubility, dissolution rate, and bioavailability of poorly water-soluble drugs, leading to the development of novel formulations (amorphous systems, microemulsions, inclusion of cyclodextrins, and solid dispersions), as well as the application of new technologies (hot melt extrusion and spray drying) [2,5,6,7,8,9].

In addition to these approaches, particle size reduction to the sub-micron size range using nanosizing techniques has been extensively explored in recent decades. Nanocrystals are crystals of pure drug with a mean diameter in the nanometer range, produced by means of top-down, bottom-up and combination approaches. Typical top-down processes require the reduction of a micronized drug powder by high pressure homogenization or wet ball milling, while in bottom-up approach drug nanocrystals are formed by precipitation [10,11]. In addition, combination approaches are a combination of a bottom-up followed by a top-down process, such as precipitation–lyophilization–homogenization technology [12].

The main advantages of nanocrystals are the simultaneous decrease in the drug particle size and the increase in its surface area, leading to an improvement of saturation solubility and drug dissolution rate [13]. This agrees with the Noyes–Whitney equation, in which the dissolution rate is a function of the surface area of the drug [14], with the Ostwald–Freundlich equation, which relates the solubility to the particle size [3], and with Kelvin’s equation, in which the increase in particle curvature improves the dissolution rate [15]. However, the application of nanocrystals dispersions has its drawbacks, since they are produced in a liquid medium, which causes their leakage from the application site. Inorganic excipients, and clay minerals in particular, which are often used in topical products to increase stability of emulsions and viscosity of suspensions, may be used to overcome this drawback [16]. In addition, they have shown advanced functionalities that make them essential ingredients in anti-inflammatory, antibacterial, and wound-healing products [17]. Both bentonite and palygorskite have recently been used to improve the bioavailability of drug candidates [18,19].

Curcumin is a well-known BCS class IV drug, characterized by antioxidant, anti-inflammatory, antibacterial and antiviral properties [20]. Among its several potential therapeutic used, curcumin has recently been proposed in the treatment of psoriatic skin lesions [21]. However, the therapeutic use of curcumin is greatly hindered by its lipophilicity and low permeability, which hampers topical bioavailability [22].

Given these premises, the aim of this work was the design and development of curcumin nanocrystals and their incorporation into natural inorganic hydrogels for topical application. In this study, curcumin nanocrystals were manufactured by the wet ball milling technique and then loaded in clay-based hydrogels. Specifically, bentonite and/or palygorskite were selected as the inorganic gelling agents. Bentonite is a swelling clay consisting of aluminum phyllosilicate mineral, while palygorskite is a magnesium aluminum silicate clay with rod and/or fibrous like crystal [23,24]. Curcumin-nanocrystal-loaded hydrogels were prepared by homogenization method and characterized, including rheological, morphological and solid-state characterization (FTIR, XRPD and thermal analysis). In vitro release, antioxidant properties and skin permeation studies were also performed.

## 2. Materials and Methods

### 2.1. Materials

Curcumin (CUR) and Kolliphor P188 (Poloxamer 188, P188) were obtained from Sigma Aldrich (Milan, Italy). Bentonite (nanoclay hydrophilic bentonite) from Sigma Aldrich (Madrid, Spain) and a pharmaceutical-grade palygorskite (Pharmasorb^®^ Colloidal) from Basf (Ludwigshafen, Germany) were used for hydrogel preparation.

All other solvents and chemicals were of high analytical quality or HPLC grade.

### 2.2. Methods

#### 2.2.1. Preparation of curcumin nanocrystals

Curcumin nanocrystals were prepared as nanosuspensions using the wet ball media milling technique. Briefly, the bulk curcumin powder was dispersed in a Poloxamer 188 water dispersion and then homogenized at 8000 rpm for 7 min using an Ultra Turrax T25 basic (IKA, Werke, Staufen, Germany). The obtained coarse suspension was then placed in conical microtubes containing about 0.4 g of 0.1–0.2 mm yttrium-stabilized zirconia–silica beads (Silibeads^®^ Typ ZY Sigmund Lindner, Warmensteinach, Germany). The microtubes were shaken for 70 min at 3000 oscillations per minute using a bead-milling cell disruptor apparatus (Disruptor Genie^®^, Scientific Industries, Bohemia, NY, USA). The obtained nanosuspensions of each microtube were gathered and then separated from the milling beads by sieving.

#### 2.2.2. Nanocrystals Characterization

The average diameter and polydispersity index (PDI, as a measure of the size distribution width) of the suspended CUR nanocrystals were estimated by Dynamic Light Scattering (DLS) using a Zetasizer nano (Malvern Instrument, Worcestershire, UK). Samples were backscattered by a helium–neon laser (633 nm) at an angle of 173° and a constant temperature of 25 °C. Zeta potential was estimated using the Zetasizer nano by means of the M3-PALS (Phase Analysis Light Scattering) technique. Immediately prior to analysis, nanosuspensions were opportunely diluted with distilled water (1:100). All the measurements were made in triplicate.

The morphology of CUR bulk powder and CUR nanocrystals was analyzed using a Zeiss ESEM EVO LS 10 (Oberkochen, Germany) environmental scanning electron microscope (ESEM), operating at 20 KV in high-vacuum modality with a secondary electron detector (SEI). For the bulk CUR analysis, the crystals were mounted on an aluminum stub with carbon adhesive discs and coated with gold in an Agar Automatic Sputter Coater B7341. For the analysis of the nanocrystals, one drop of CUR nanosuspension was firstly placed on a glass slide and air-dried, and then mounted on the stub following the procedure stated above.

#### 2.2.3. Preparation of Curcumin-Nanocrystal-Loaded Hydrogels

Pure hydrogels were prepared by weighing into a screw-capped sample vial the amount of bentonite (30% *w*/*w*), palygorskite (30% *w*/*w*) or bentonite/palygorskite (15% *w/w* for each clay) in distilled water. Bentonite water gels require a careful preparation procedure, as mixing conditions greatly influence the resulting dispersions [25]. Similarly, palygorskite is a tubular natural excipient able to form three-dimensional structures (gels) composed of interconnecting fibers once clay aggregates have been fully separated by high mixing energies. Consequently, the clay/water mixtures were homogenized by means of a turbine high-speed agitator (Silverson LT, London, UK) for 10 min at 24,000 rpm and left at room temperature until the hydrogels were obtained: bentonite-based hydrogel (B), palygorskite-based hydrogel (P), and bentonite and palygorskite-based hydrogel (B/P).

As for the CUR nanocrystal-loaded hydrogels, CUR nanocrystals were mixed in a 1:1 weight ratio with B, P and B/P hydrogels to obtain a final curcumin concentration of 0.5% *w*/*w*. The mixture was homogenized as previously reported until curcumin nanocrystal-loaded hydrogels (B-CUR, P-CUR, or B/P-CUR) were obtained.

#### 2.2.4. Rheological Properties

Rheological measurements were performed by means of a controlled-rate viscometer (Thermo Scientific HAAKE, RotoVisco 1, Karlsruhe, Germany) equipped with a plate/plate combination (Ø 20 mm serrated PP20S sensor system). The temperature was set at 32 (±0.5) °C by means of a Peltier temperature controller. The rheological properties of hydrogel samples were measured within the shear rate range of 10–500 s^−1^. The shear stress vs. shear rate dependencies measured at increasing and then immediately decreasing shear rate were used to determine the thixotropic properties.

#### 2.2.5. Solid State Characterization

Solid state characterization of the hydrogels was performed after freeze drying. For this purpose, the samples were placed into a freezer (−80 °C) and then put into a freeze-dryer (CoolSafe Touch 100-9, LaboGene, Aarhus, Denmark) operating under low vacuum (below 4 hPa) for 24 h.

The surface morphology of the freeze-dried hydrogels was also assessed by Scanning Electron Microscopy (SEM). The images were taken on a Hitachi S-510 microscope (Hitachi Scientific Instruments Ltd., Tokyo, Japan) operating at 25 kV in low-vacuum mode.

X-ray powder diffraction (XRPD) was carried out using an X-Pert Pro^®^ diffractometer (Marvel Panalytical, Madrid, Spain) with the CuKα radiation (λ = 1.5405 Å). The powder samples were scanned in a 2θ angle range of 4°–70°. The diffraction results were analyzed with the X’Pert HighScore^®^ software (Version 2.2., Marvel Panalytical, Madrid, Spain).

Fourier transform infrared spectroscopy (FTIR) spectra were recorded on a 6200 spectrophotometer (JASCO, Pfungstadt, Germany), equipped with a software Spectra Manager (version 2, JASCO, Pfungstadt, Germany) and an attenuated total reflectance accessory. Measurements were carried out from 400 to 4000 cm^−1^ at 2 cm^−1^ resolution.

Thermogravimetric analysis (TGA) and differential scanning calorimetry (DSC) were performed using a model TGA/DSC1 calorimeter (Mettler Toledo, Madrid, Spain) equipped with a sensor and FRS5 microbalance (precision 0.1 μg). Samples (40 mg approximately) were heated in air atmosphere at 10 °C/min in 30–935 °C temperature range.

#### 2.2.6. In Vitro Release Study of Curcumin-Nanocrystal-Loaded Hydrogels

The release studies were performed in a mixture of phosphate buffer solution pH 7.4 (PBS) and ethanol (1:1 *v*/*v*) at 32 °C using the dialysis bag technique. Then, 100 mg of hydrogel was placed into dialysis bags (MWCO: 12000–14000 Da, Medicell Membranes Ltd., London, UK), filled with 100 µL of PBS, closed with fluorocarbon yarn and laid in stainless steel sinkers. Then, the bags were soaked in falcon tubes containing 40 mL of PBS. The tubes were transferred into a shaking water bath, kept at a temperature of 32 °C and shaken at a constant speed of 50 rpm. At prefixed times, 1 mL of dissolution medium was withdrawn, and the same volume was replaced to maintain the sink conditions.

The amount of CUR released was assessed by means of HPLC analysis, using an Agilent HPLC-UV/DAD apparatus (Agilent Infinity II; Agilent Technologies Inc., Waldbronn, Germany) and a Sunfire C18 column (3.5 µm × 4.6 mm × 150 mm) (Waters, Milford, MA, USA) as stationary phase. The mobile phase was a mixture of acetonitrile, water, and acetic acid (95: 4.84: 0.16 *v*/*v*). The detection wavelength was set at 421 nm, the injection volume was 10 μL, the flow rate was 0.5 mL/min and the temperature of column oven was set at 25 °C.

#### 2.2.7. Antioxidant Properties

The supernatants corresponding to 100% curcumin release obtained from the in vitro release experiments were subjected to DPPH assay. Each sample was mixed 1:1 weight ratio with DPPH (Sigma-Aldrich, Madrid, Spain) methanol solution (8 μg/mL), kept for 30 min in the dark, and the absorbance was measured at 515 nm (Lamba 35, Perkin Elmer, Madrid, Spain). The results are expressed as radical scavenging activity percentage, as follows:RSA% = 1 − ((A_sample_ − A_blank_)/A_control_)
where A_sample_ is the absorbance of the sample after 30 min of incubation with DPPH, A_blank_ is the absorbance of the sample before the reaction and A_control_ the absorbance of DPPH.

#### 2.2.8. In Vitro Skin Permeation Study

In vitro permeation experiments were performed by means of Franz-type diffusion cells (Perme Gear, Pensilvania USA) with an effective diffusion area of 0.75 cm^2^, using ear skin excised from pigs provided by a local slaughterhouse (Granada, Spain). After careful removal of the subcutaneous fat, the skin was cut into squares of 3 × 3 cm and stored at −20 °C until use. To increase reproducibility, skin was pre-hydrated with PBS for 30 min before starting experiments. The skin was positioned with the stratum corneum facing the donor phase, between the donor and the receptor compartments of the Franz cell. The receptor chamber was filled with a mixture of PBS (pH 7.4) and ethanol (1:1 *v*/*v*) previously degassed and continuously stirred by a magnetic bar. Each hydrogel (100 mg) was placed onto the skin, thermostated at 37 °C to obtain the physiological skin temperature (32 °C). After 24 h, the skin surface was washed, and the stratum corneum (SC) was separated by stripping with adhesive tape (Strappal^®^, Barcelona, Spain). The adhesive tape was firmly pressed on the SC and pulled off with a rapid and fluent stroke. The epidermis was separated from the dermis with a surgical scalpel. Tape strips, epidermis and dermis were put individually in methanol [26]. The resulting samples were sonicated to extract CUR, and the methanol extracts were analyzed by HPLC.

#### 2.2.9. Statistical Analysis

Statistical analysis was performed using Astatsa statistical calculator. One-way analysis of variance (ANOVA) was followed by Scheffè for post hoc comparisons. *p* < 0.05 was considered significant.

## 3. Results and Discussion

### 3.1. Nanocrystals Characterization

In this paper, CUR nanocrystals were prepared as nanosuspensions by the wet ball media milling technique using specific amounts of the different components (1% of CUR, 0.5% of Poloxamer 188 and 98.5% *w*/*w* of water). In a previous study, the concentration of Poloxamer 188, used as stabilizer, and the milling time (70 min) were optimized in order to obtain a 1% CUR nanosuspension formulation with small average diameter and PDI [27]. The composition and characterization of the prepared CUR nanosuspension is reported in Table 1.

On the day of preparation, the CUR nanosuspension showed a mean diameter of 203 nm and a polydispersity index (PDI) of 0.24, demonstrating a relatively narrow size distribution. The zeta potential value, determined by means of the M3-PALS technique, was relative weakly negative (−21.3 mV). However, a previous study revealed the long-term stability of the nanosuspension formulation.

The study of the morphology of bulk CUR crystals and CUR nanocrystals have been carried out by environmental scanning electron microscope (Figure 1). As can be seen from the micrographs, the crystals of the raw CUR appear large with irregular elongated shape (Figure 1a). Conversely, after the milling, the CUR nanocrystals show a regular and rounded shape, with a homogenous particle size distribution, in accordance with DLS analysis (Figure 1b).

### 3.2. Rheological Analysis

In Figure 2 (top panel), the viscosity profiles of B-CUR, P-CUR, and B/P-CUR hydrogels are reported.

B-CUR hydrogel was characterized by a non-Newtonian behavior due to a reduction in viscosity with increasing shear rate. P-CUR hydrogel was also characterized by a pseudoplastic behavior, and the viscosity values at low shear rate were 5-fold higher than that of B-CUR. When both palygorskite and bentonite were present, B/P-CUR hydrogel exhibited a similar profile, although the viscosity values were about 3-fold lower than P-CUR hydrogel. This difference could be explained as being the result of multiple factors. In the suspension of pure palygorskite, the individual rods or small crystal bundles are in a random orientation and associated with each other to form clusters. A lot of water molecules are entrapped in these clusters, resulting in an increase of the effective solid volume fraction of the suspension and a higher viscosity. Therefore, the decrease of viscosity for B/P-CUR could be attributed to the reduction of interparticle repulsive force and the collapse of network structure.

Figure 2 (bottom panel) shows the typical experimental dependences of the shear stress vs. shear rate recorded with increasing and subsequently decreasing shear rate.

All the hydrogels formed hysteresis loops, characteristic of the thixotropic materials. This indicates that the hydrogels exhibited two kinetic processes: the structure of the hydrogel is broken down by shear (constant or increasing) applied, and then the structure is rebuilt after the shear stress is removed. The thixotropy of both bentonite and palygorskite hydrogels has been related to the agglomerated (coagulated) state of the clay, due to attractive particle–particle interactions. For example, palygorskite suspensions are antithixotropic when they are in a dispersed state and thixotropic when they are coagulated [28]. In addition, it has been reported that the thixotropy of bentonite-based suspensions is proportionally related to the concentration of the bentonite particles [29]. All of these factors may enhance the occurrence of thixotropy.

In topical application, low resistance (low viscosity) to shaking is expected, facilitating administration over the skin, and after use, the product quickly returns to its original form as a gel to prevent segregation of the components.

### 3.3. Solid State Characterization

In Figure 3, the SEM microphotographs of freeze-dried B-CUR, P-CUR, and B/P-CUR hydrogels are presented. The morphologies are highly affected by the presence of different clays. B-CUR hydrogels were characterized by porous structure that resulted from the sublimation of water during freeze drying, with a flake-like structure. On the other hand, P-CUR hydrogel showed a fiber-like structure with many individual needle-like crystals of varying dimensions, typical of palygorskite. Finally, B/P hydrogel exhibited the typical features of both bentonite and palygorskite.

The crystalline structure of the hydrogels was studied by XRPD analysis, as reported in Figure 4 (B-CUR—top panel; P-CUR–middle panel, and B/P-CUR—bottom panel).

The hydrogel diffractograms show the reflections ascribable to the pure bentonite and palygorskite. For bentonite-based hydrogels, diffraction peaks are noticeable at 5.8° and 20° 2θ, while the typical diffraction peaks at 8.5°, 21.0° and 26.7° 2θ are evident in palygorskite-based hydrogels [30,31]. As reported by Casula et al. [27], CUR nanocrystals showed a crystalline profile due to the high crystallinity of both pristine curcumin and Poloxamer 188, used for nanosuspension preparation. However, these crystalline peaks are completely overlapped in the hydrogels’ spectra, which are dominated by the presence of the clays.

Figure 5 reports FTIR spectra of the pure components and CUR nanocrystal-loaded hydrogels. Regarding bentonite, the vibrational band at 3622 cm^−1^ is related to octahedral layer-OH groups (Al-OH-Al, Al-OH-Mg and Mg-OH-Mg vibrations) typical of clay minerals [30]. Moreover, a broad band around 3400 cm^−1^ is due to OH stretching of water molecules situated in the interlayer space of the clay. In addition, the coordination of bentonite exchangeable cations to water molecules led to an intense band al 1632 cm^−1^. The bands at 1100 cm^−1^ and 985 cm^−1^ can be attributed to Si-O-Si out-of-plane and in-plane bending, respectively.

The palygorskite spectrum is characterized by bands at 3616 cm^−1^ and 3542 cm^−1^ due to the OH stretching vibrations in the Si–OH bond. The band at 1652 cm^−1^ was related to bending of absorbed and zeolitic water, while characteristic bands of silicate were evident in the range between 1200 and 400 cm^−1^.

The spectrum of CUR nanocrystals highlighted the same peaks of both pristine curcumin and Poloxamer 188. In detail, two broad peaks at 3326 cm^−1^ and at 2881^−1^ are related to phenolic OH stretching and C–H stretching vibration, respectively.

The hydrogels’ spectra are dominated by the presence of clay except for the vibrations at 1602 and 1510 cm^−1^ related to C=C and C=O stretching of CUR. Moreover, it is conceivable that there is no strong chemical interaction between curcumin and clay, since no significant band shifts were observed in the FTIR spectra.

Figure 6 shows TGA spectra of curcumin nanocrystal-loaded hydrogels (B-CUR, P-CUR, and B/P-CUR).

The TGA thermogram of pure bentonite revealed an initial slight weight loss, followed by a dehydroxylation step in the range temperature between 600 °C and 700 °C [24]. Pristine palygorskite exhibited four stages of mass loss: Firstly, a weight loss of 8% occurs at 95 °C, corresponding to surface water loss. The second weight loss of 3% at 220 °C is related to loss of zeolitic water located in the channels. The third thermal event corresponds to the water linked to the octahedral ions (weight loss of 4%) at 440 °C and, finally, a mass loss of 1.89% is due to the dehydroxylation of the clay at 640 °C. DSC thermograms of both bentonite and palygorskite confirmed their stability in the temperature range evaluated. Moreover, endothermic degradation peaks of CUR nanocrystals were measured over 400 °C in the studied inorganic gels, confirming the stable effect of the formulations, as curcumin degradation has been reported to occur at lower temperature [27].

CUR-loaded hydrogels did not show melting peaks or endothermic events related to curcumin decomposition, suggesting that the hydrogels were able to protect the curcumin nanocrystals from thermal degradation.

### 3.4. In Vitro Release Study of CUR Nanocrystal-Loaded Hydrogels

Figure 7 reports the release profiles of curcumin from B-CUR, P-CUR, and B/P-CUR hydrogels.

The release profiles consist of two steps: an initial step in which a steep increase of CUR released is noticeable within the first 8 h, independently of the clay type, and a second stage characterized by a decrease in the release. The two behaviors could be related to the hydration, swelling and degradation of the systems. In particular, the burst release could be associated with the hydration and a partial loss of integrity, causing a fast release of the CUR present in the surface layers. The second step corresponds to the slower hydration of the hydrogel inner part and the consequent diffusion of CUR nanocrystals across the hydrogel network.

### 3.5. Antioxidant Properties

DPPH radical with a characteristic absorption at 517–520 nm was used to evaluate the antioxidant activity of CUR. When an antioxidant encounters the DPPH, a stable free radical is formed and scavenged, leading to the reduction in absorbance. Figure 8 shows the antioxidant activity of the hydrogel-bead-loaded Cur. CUR-loaded hydrogels exhibited antioxidant properties, reaching a radical scavenging activity of 78% in both B-CUR and B/P-CUR hydrogels.

On the other hand, a decrease in the scavenging activity was noticed when curcumin nanocrystals were loaded in palygorskite-based hydrogels, probably due to their higher viscosity and slower release of curcumin. In addition, this behavior could also be related to curcumin diffusion inside the nanotunnels of palygorskite without steric impediment due to the proper size match between curcumin molecules and palygorskite nanochannels. The ROS scavenging properties of curcumin-loaded hydrogels could be attributed to the protection of curcumin nanocrystals in the hydrogel form, which could act as classic phenolic compound by donating hydrogen atoms from its phenolic groups [32].

### 3.6. In Vitro Skin Permeation Study

Figure 9 reports curcumin skin penetration into stratum corneum (SC), epidermis (E), dermis (D) and receptor chamber (CR) from B-CUR, P-CUR, and B/P-CUR hydrogels. The skin permeation studies were carried out using 1:1 (*v*/*v*) PBS/ethanol mixture as receptor medium, which is considered in the literature to be a suitable release medium [33,34,35], wherein curcumin is soluble.

Independently of clay type and concentration, all hydrogels lead to an accumulation of CUR in the upper skin layers (stratum corneum and epidermis). With respect to the deeper skin layers, drug penetration revealed an inverse correlation between hydrogel viscosity and curcumin penetration. Among hydrogels, the highest curcumin amount was delivered to the deeper skin layers by the B-CUR and B/P-CUR hydrogels, while the smallest amount of curcumin was found upon the application of P-CUR hydrogel.

On the basis of all results, the addition of bentonite to the colloidal suspension in both B-CUR and B/P-CUR hydrogels provided an increase of CUR skin penetration in the receptor chamber, probably due to their desirable rheological characteristics, which should guarantee the necessary contact time of the gel with the skin and also high stability. Therefore, these gels were considered to be the optimal formulation of curcumin. On the other hand, the P-CUR hydrogel led to an increase of CUR concentration in the skin dermis compared to the other hydrogels. In addition, the lower CUR accumulation of P-CUR hydrogel in the deeper skin layers could be related to a decrease in nanocrystal mobility due to the increase in the viscosity of the dispersant phase [36].

According to Koop et al. [37], CUR preferentially accumulates in the stratum corneum rather than in the epidermis and dermis due to the higher affinity of the drug to the superficial layers of the skin. Moreover, lipid layers and follicles seem to be the preferred pathways for the penetration of lipophilic drugs, and nanocarriers could modulate the delivery of drugs and residence time in the skin [38].

Although it is conceivable that curcumin is released more rapidly from nanocrystals than when loaded into clay-based hydrogels, its application in the form of nanocrystals is not feasible, as they are in an aqueous solution consisting of 98.5% water, which would cause leakage from the application site, resulting in poor drug absorption. To this end, clay minerals have been used to overcome this drawback, leading to an increase in the stability and viscosity of nanocrystals and improving curcumin bioavailability.

## 4. Conclusions

Clay-based hydrogels loaded with curcumin nanocrystals were successfully designed and developed for topical application. Curcumin nanosuspension was prepared using the wet ball media milling technique, showing a mean diameter of 203 nm and a weakly negative zeta potential. CUR nanocrystals were then loaded in clay-based hydrogels by means of homogenization method. The hydrogels were characterized by a crystalline structure and did not show endothermic events related to curcumin decomposition indicating that the hydrogels were able to protect curcumin nanocrystals from thermal degradation. Moreover, the physico-chemical characterization suggested that no strong chemical interaction between curcumin nanocrystals and clay occurred, and bentonite and/or palygorskite played a crucial role in the rheological behavior of the formulation and consequently in curcumin nanocrystals release. Drug penetration exhibited an inverse correlation between hydrogel viscosity and curcumin penetration. In particular, the presence of bentonite provided an increase of curcumin skin penetration and simultaneously allowed its radical scavenging properties, due to the desirable rheological characteristics, which should guarantee the necessary contact time of the gel with the skin. In conclusion, bentonite-based hydrogels showed superior properties from the point of view of both physico-chemical and in vitro studies, representing a promising formulation for topical delivery of curcumin.

## Figures and Tables

**Figure 1 pharmaceutics-14-02836-f001:**
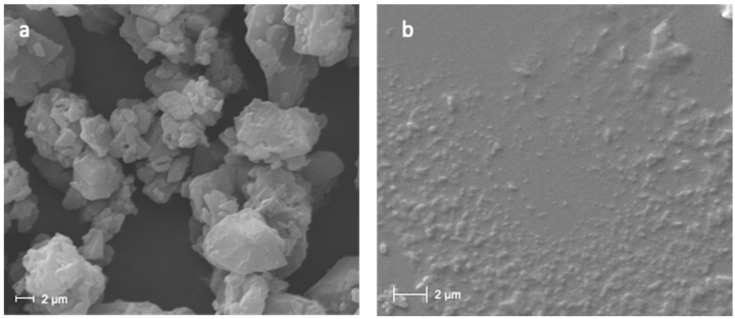
ESEM micrographs of CUR bulk powder (**a**) and CUR Nanocrystals (**b**).

**Figure 2 pharmaceutics-14-02836-f002:**
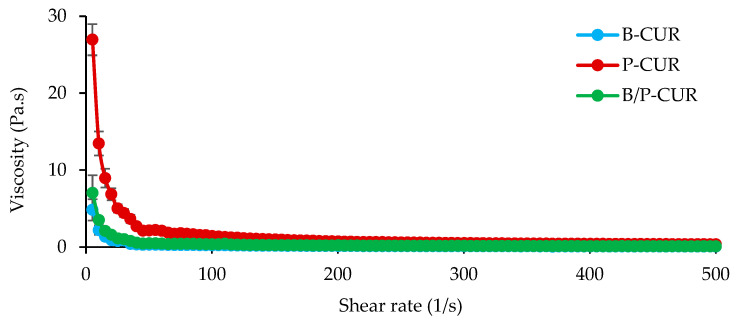
Viscosity profiles (top panel) and flow curves (bottom panel) of B-CUR, P-CUR, and B/P-CUR hydrogels measured up to 500 s^−1^ at 32 °C (mean values ± s.d.; *n* = 3).

**Figure 3 pharmaceutics-14-02836-f003:**
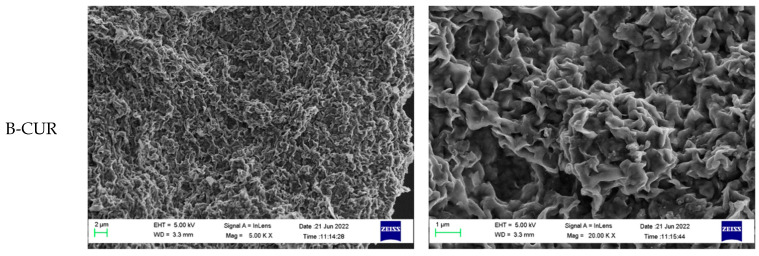
SEM microphotographs of freeze-dried B-CUR (top panel), P-CUR (middle panel), and B/P-CUR (bottom panel) hydrogels.

**Figure 4 pharmaceutics-14-02836-f004:**
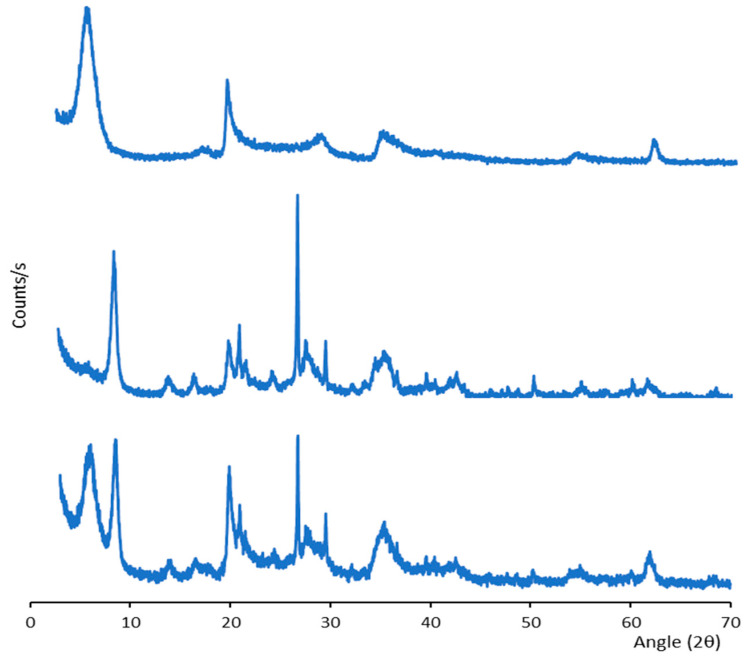
XRPD diffraction patterns of B-CUR (top panel), P-CUR (middle panel), and B/P-CUR (bottom panel) hydrogels.

**Figure 5 pharmaceutics-14-02836-f005:**
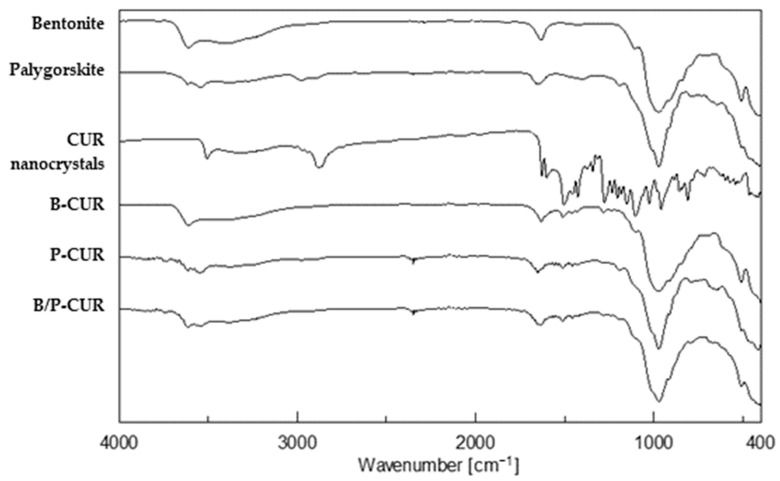
FTIR spectra of the pure components (bentonite, palygorskite, and curcumin nanocrystals) and hydrogels (B-CUR, P-CUR, and B/P-CUR).

**Figure 6 pharmaceutics-14-02836-f006:**
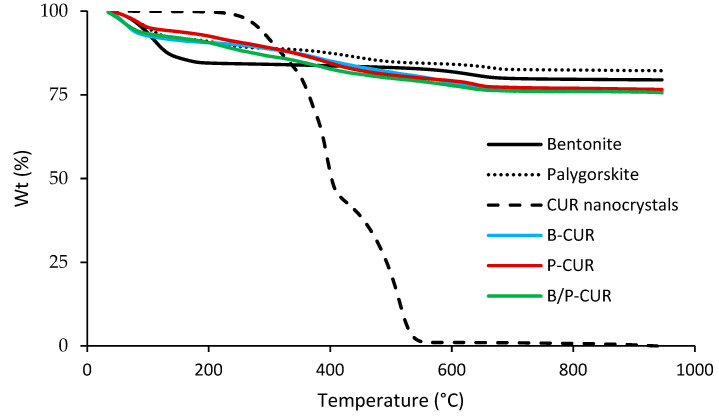
TGA curves of pristine components and curcumin nanocrystal-loaded hydrogels (B-CUR, P-CUR, and B/P-CUR).

**Figure 7 pharmaceutics-14-02836-f007:**
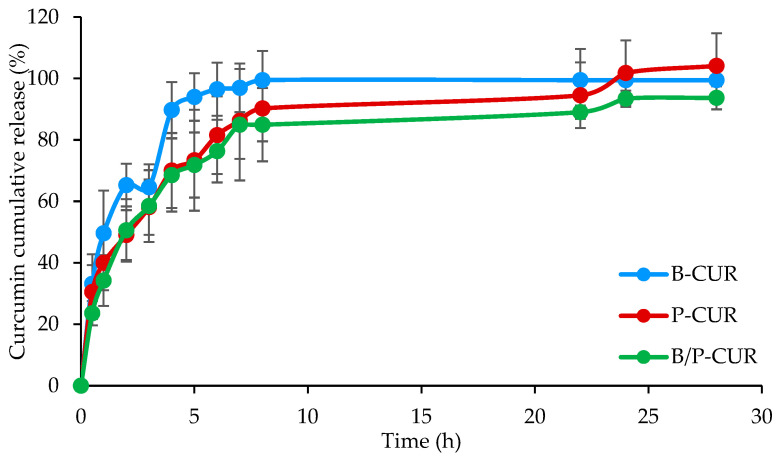
In vitro cumulative release profiles of curcumin from B-CUR, P-CUR, and B/P-CUR hydrogels (mean values ± s.d.; *n* = 3). ANOVA one-way; Scheffe’s test (*p* ≤ 0.05): B-CUR 2 h vs. P-CUR 2 h.

**Figure 8 pharmaceutics-14-02836-f008:**
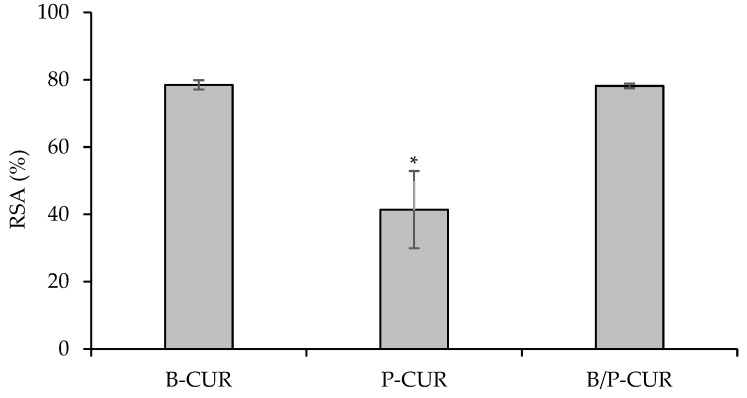
Radical scavenging activity (RSA%) of curcumin released from B-CUR, P-CUR, and B/P-CUR hydrogels (mean values ± s.d.; *n* = 3). * indicates significant differences (*p* < 0.05).

**Figure 9 pharmaceutics-14-02836-f009:**
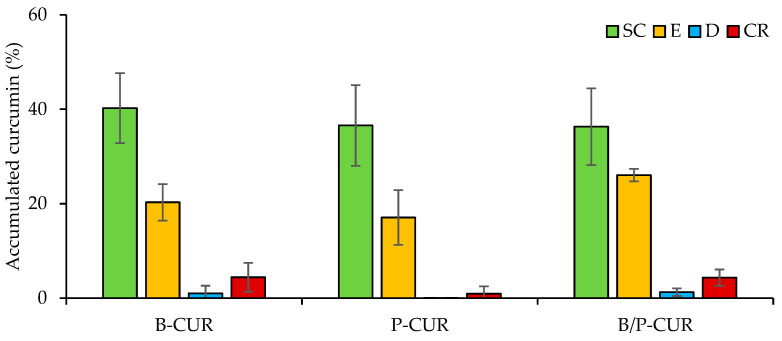
Cumulative amount of curcumin retained in and permeated through the skin layers 24 h after the application of B-CUR, P-CUR, and B/P-CUR hydrogels. The amount of curcumin is expressed as the percentage of the dose applied on the skin (mean values ± s.d.; *n* = 3).

**Table 1 pharmaceutics-14-02836-t001:** Composition and characterization of CUR nanosuspension at the day of preparation.

CUR Nanosuspension Composition		Characterization		
Component	% (*w*/*w*)	Mean Diameter (nm)	PDI	Zeta Potential (mV)
Curcumin	1.0	203 ± 7	0.24 ± 0.02	−21.3 ± 1.3
Poloxamer 188	0.5			
Water	98.5			

## Data Availability

Data available on request.

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
