# Peer review of "Clay-Based Hydrogels as Drug Delivery Vehicles of Curcumin Nanocrystals for Topical Application"

_pharmaceutics, 2022, doi:10.3390/pharmaceutics14122836_

Round 1

Reviewer 1 Report

The manuscript numbered pharmaceutics-2081966 by M. Ruggeri et al. describes the preparation of several pharmaceutical forms of curcumin. The curcumin forms obtained were characterized physicochemically and the release profile, free radical scavenging capacity and skin application were checked. The results obtained are predictable prior to the study. 

I have the following comments on the manuscript:

1) The abbreviation "BCS" was used in the abstract (l. 14), which was not explained. It is admittedly explained in the Introduction, but the abstract must be understood without referring to the entire manuscript. It is not a commonly used abbreviation, such as API or NMR, so it should be explained.

2) A sentence in the abstract (l. 19-22) is too similar (beginning the same) to another sentence in the Introduction (l. 81-84). These sentences incorrectly use the term "a multidisciplinary approach." This should be corrected.

3) The first sentence in the Introduction (l. 29-32) talks about recent research in pharmaceutical science, while the literature reference used to support this view is from 2009. 

4) (l. 32 and 33) Since the Authors often introduce abbreviations and explain them, it would also be useful to introduce the abbreviation "API", since it is used later. 

5) The abbreviation "Q3C" (l. 41) is not explained.

6) There is no reason to write the word "curcumin" with a capital letter (l. 72 and elsewhere), especially since the Authors mostly use lowercase spelling of this substance.

7) The self-citations (Refs. 23 and 24) are quite general, I would replace them with works by other researchers.

8) The title of Paragraph 2.2.1 uses the abbreviation "CUR" for the first time, which is not explained anywhere. In my opinion, there is no need to abbreviate the name "curcumin". The full name and the abbreviation appear interchangeably. One notation should be used consistently. 

9) Paragraph 2.2.3 (l. 128, Refs. 25-27) unnecessarily uses as many as three works of their own. One example would have sufficed.

10) The abbreviations B, P and B/P are used for the first time in l. 133. The meaning of these abbreviations should be explained, although one can guess.

11) Software needs references (l. 157, 159 and 160). Missing here.

12) Are you sure the FTIR spectra were recorded with a resolution of 0.25 cm-1? This seems unlikely to me. The most common resolution is 4 cm-1

 13) What does "a mod." mean. in l. 163 regarding the calorimeter? Usually a few to several mg of substance is used in calorimeters. Here it says that the samples had a mass of about 40 mg.

14) Since the abbreviation "w/w" was used earlier, this designation should be used instead of "weight ratio" in l. 169. This is a very common abbreviation, so there is no need to explain it.

15) I believe that the self-citation of Ref. 28 (l. 192) is completely unnecessary.

16) l. 202 "1:1". Is this about the volume ratio? It should be written.

17) The "Results" paragraph occurs without the "Discussion" paragraph. Therefore, this part of the paper should be called "Results and Discussion".

18) Table 1 incorrectly states the values with uncertainties. The uncertainty must have the same decimal expansion as the result. Thus, the notation 203.10+/-7.3 is incorrect. In addition, the uncertainty can have up to two significant digits, so the uncertainty "1.32" is not rounded.

19) "XRD" (l. 279). This abbreviation is used here not in accordance with other abbreviations used for X-ray powder diffraction. On the next page, the title of Figure 4 is "XRPD."

20) (l. 299) The term "bending" is missing for the words "out-of-plane and in-plane."

21) The name "poloxamer 188" in line 305 is lowercase while elsewhere the spelling is uppercase.

22) Repetition of the word "reported" in two sentences side by side (l. 314-318). This needs to be corrected.

23) Comparing or giving values in percentages is used for general comparison, so expanding to two decimal places the percentages, in my opinion, is not justified. 8.34% (l. 319); 2.88% (l. 320); and 4.40% (l. 321).

24) The title of Figure 6. I would use the term "TGA curves" rather than "TGA spectra." The latter is more appropriate for spectroscopy.

25) Biological material was used in the study and there is no mention of bioethics committee approval. 

25) Roughly one-third of all references are self-citations (11 out of 35). One should avoid citing one's own work unless clearly justified, the lack of research on a given topic by other researchers. I have listed above the self-citations that, in my opinion, are not justified.

Author Response

The manuscript numbered pharmaceutics-2081966 by M. Ruggeri et al. describes the preparation of several pharmaceutical forms of curcumin. The curcumin forms obtained were characterized physicochemically and the release profile, free radical scavenging capacity and skin application were checked. The results obtained are predictable prior to the study.

The authors wish to thank the reviewer for his/her fruitful comments to improve the paper quality. Please note that in the text all the corrections have been underlined.

I have the following comments on the manuscript:

1) The abbreviation "BCS" was used in the abstract (l. 14), which was not explained. It is admittedly explained in the Introduction, but the abstract must be understood without referring to the entire manuscript. It is not a commonly used abbreviation, such as API or NMR, so it should be explained.

The abbreviation was written in full in the abstract section: Curcumin is a well-known Biopharmaceutics Classification System (BCS) class IV drug

2) A sentence in the abstract (l. 19-22) is too similar (beginning the same) to another sentence in the Introduction (l. 81-84). These sentences incorrectly use the term "a multidisciplinary approach." This should be corrected.

The sentence has been changed in the abstract as follow: Curcumin nanocrystals-loaded hydrogels were manufactured by means of a homogenization process and characterized for their chemico-physical properties, in vitro release, antioxidant activity and skin permeation.

Moreover, the term "a multidisciplinary approach." Has been checked.

3) The first sentence in the Introduction (l. 29-32) talks about recent research in pharmaceutical science, while the literature reference used to support this view is from 2009.

The reference has been replaced with Adity, S.; Karan, M.; Daisy, A.; Subrahmanya, G.S, A Comprehensive Review on Strategies for New Drug Discovery and Enhanced Productivity in Research and Development: Recent Advancements and Future Prospectives. Mini-Rev Org Chem 2021, 18, 361-382.

4) (l. 32 and 33) Since the Authors often introduce abbreviations and explain them, it would also be useful to introduce the abbreviation "API", since it is used later.

The abbreviation API has been explained in the abstract: The poorly water-solubility of significant number of active pharmaceutical ingredients (API) remains one of the main challenges in the drug development process, causing low bioavailability and therapeutic failure of drug candidates

5) The abbreviation "Q3C" (l. 41) is not explained.4

Q3C is not an abbreviation, but the guideline number. The sentence has been modified as follows: However, the International Council for Harmonisation (ICH) Q3C guidelines on impurities (guidelines for residual solvents) limits the use of solvents in pharmaceutical applications.

6) There is no reason to write the word "curcumin" with a capital letter (l. 72 and elsewhere), especially since the Authors mostly use lowercase spelling of this substance.

The capital letter has been removed

8) The title of Paragraph 2.2.1 uses the abbreviation "CUR" for the first time, which is not explained anywhere. In my opinion, there is no need to abbreviate the name "curcumin". The full name and the abbreviation appear interchangeably. One notation should be used consistently.

The abbreviation CUR has been removed from the paragraph titles. However, the abbreviation was introduced in the material section and used throughout the text and figures

10) The abbreviations B, P and B/P are used for the first time in l. 133. The meaning of these abbreviations should be explained, although one can guess.

The sentence has been modified as follows: Consequently, the clay/water mixtures were homogenized by means of a turbine high-speed agitator (Silverson LT, United Kingdom) for 10 min at 24000 rpm and left at room temperature until the hydrogels were obtained: bentonite-based hydrogel (B), palygorskite-based hydrogel (P), and bentonite and palygorskite-based hydrogel (B/P).

11) Software needs references (l. 157, 159 and 160). Missing here.

The software references have been added

12) Are you sure the FTIR spectra were recorded with a resolution of 0.25 cm-1? This seems unlikely to me. The most common resolution is 4 cm-1.

FTIR spectra were recorded with a resolution of 2 cm-1  and it has been checked in the method section.

 13) What does "a mod." mean. in l. 163 regarding the calorimeter? Usually a few to several mg of substance is used in calorimeters. Here it says that the samples had a mass of about 40 mg.

The word mod. has been changed with model. Moreover, the authors confirm that about 40 mg of sample were used.

14) Since the abbreviation "w/w" was used earlier, this designation should be used instead of "weight ratio" in l. 169. This is a very common abbreviation, so there is no need to explain it.

16) l. 202 "1:1". Is this about the volume ratio? It should be written.

Lines 168 and 201: The word 1:1 v/v has been added

17) The "Results" paragraph occurs without the "Discussion" paragraph. Therefore, this part of the paper should be called "Results and Discussion".

The paragraph title has been checked (l. 215)

18) Table 1 incorrectly states the values with uncertainties. The uncertainty must have the same decimal expansion as the result. Thus, the notation 203.10+/-7.3 is incorrect. In addition, the uncertainty can have up to two significant digits, so the uncertainty "1.32" is not rounded.

The table has been corrected: all the values have 2 decimal digits

19) "XRD" (l. 279). This abbreviation is used here not in accordance with other abbreviations used for X-ray powder diffraction. On the next page, the title of Figure 4 is "XRPD."

The “XRD” has been changed with ”XRPD”, in accordance with other abbreviations

20) (l. 299) The term "bending" is missing for the words "out-of-plane and in-plane."

The term "bending" has been added

21) The name "poloxamer 188" in line 305 is lowercase while elsewhere the spelling is uppercase.

The name "poloxamer 188" has been checked

22) Repetition of the word "reported" in two sentences side by side (l. 314-318). This needs to be corrected.

The sentence has been modified as follows: Figure 6 shows TGA spectra of curcumin nanocrystals-loaded hydrogels (B-CUR, P-CUR, and B/P-CUR).

23) Comparing or giving values in percentages is used for general comparison, so expanding to two decimal places the percentages, in my opinion, is not justified. 8.34% (l. 319); 2.88% (l. 320); and 4.40% (l. 321).

The percentage values have been checked and reported without decimal digits

24) The title of Figure 6. I would use the term "TGA curves" rather than "TGA spectra." The latter is more appropriate for spectroscopy.

The term TGA curves has been added

25) Biological material was used in the study and there is no mention of bioethics committee approval.

A sentence has been added in Institutional Review Board Statement section: Ethical review and approval were waived for this study due to the use of skin samples provided from a local slaughterhouse and obtained from animals that died of natural causes.

7) The self-citations (Refs. 23 and 24) are quite general, I would replace them with works by other researchers.

9) Paragraph 2.2.3 (l. 128, Refs. 25-27) unnecessarily uses as many as three works of their own. One example would have sufficed.

15) I believe that the self-citation of Ref. 28 (l. 192) is completely unnecessary.

25) Roughly one-third of all references are self-citations (11 out of 35). One should avoid citing one's own work unless clearly justified, the lack of research on a given topic by other researchers. I have listed above the self-citations that, in my opinion, are not justified.

 The references 23 and 24 have been replaced with works of other researchers.

References 26-27 have been removed

Reference 28 has been removed,

5 self-citations have been removed and replaced with other references.

Reviewer 2 Report

-Ethanol in the receptor compartment might influence skin permeability in a bio-irrelevant manner. Why did the authors chose this way for creating sink conditions?

-Why did not the authors attempt characterizing potential thixotropic behavior of the system?

-Lack of control for drug release and permeation studies: This is a serious shortcoming. A control with the nanocrystals only without clay additives needs to be run.

-For TGA and XRPD experiments: curcumin nanocrystals as well as the clays need to be run separately as well as a physical mixture ( simple powder-powder mixing)

Author Response

The authors wish to thank the reviewer for his/her fruitful comments to improve the paper quality. Please note that in the text all the corrections have been underlined.

1) Ethanol in the receptor compartment might influence skin permeability in a bio-irrelevant manner. Why did the authors chose this way for creating sink conditions?

A PBS/ethanol mixture was chosen based on the sink conditions. In fact, the solubility of curcumin nanocrystals in the PBS/ethanol mixture was about 10 folds higher than the solubility in PBS. Therefore, the maximum amount of curcumin-loaded hydrogels was about 10 mg considering PBS in the receptor chamber, and this was not achievable. Moreover, the use of PBS/ethanol mixture is recurrent in literature and a sentence has been added in the results and discussion section:

The sink permeation studies were carried out using 1:1 (v/v) PBS/ethanol mixture as receptor medium, considered in the literature as a suitable release medium [33-35], wherein curcumin is soluble.

2) Why did not the authors attempt characterizing potential thixotropic behavior of the system?

The thixotropic behavior of the hydrogels has been evaluated. A sentence in the method section and a comment in results and discussion have been added. Figure 2 (bottom panel) shows the typical experimental dependences of the shear stress vs. shear rate recorded with increasing and subsequently decreasing shear rate. All the hydrogels formed hysteresis loops, characteristic of the thixotropic materials. This indicates that the hydrogels exhibited two kinetic processes: the structure of the hydrogel is broken down by shear (constant or increasing) applied, and then the struc-ture is rebuilt after the shear stress is removed. Thixotropy of both bentonite and palygorskite hydrogels has been related to an agglomerated (coagulated) state of the clay due to attractive particle-particle interac-tions. For example, palygorskite suspensions are antithixotropic when they are in a dispersed state, and thixotropic when they are coagulated [28]. In addition, it has been reported that the thixotropy of bentonite-based suspension is proportionally related to concentration of the bentonite particles [29]. All of these factors may enhance the oc-currence of thixotropy. In topical application, low resistance (low viscosity) to shaking is expected, facili-tating administration over the skin, and after use, the product quickly returns to its original form as a gel to prevent segregation of the components. 

Figure 2. Viscosity profiles (top panel) and flow curves (bottom panel) of B-CUR, P-CUR, and B/P-CUR hydrogels measured up to 500 s−1 at 32 °C (mean values±s.d.; n=3).

3) Lack of control for drug release and permeation studies: This is a serious shortcoming. A control with the nanocrystals only without clay additives needs to be run.

A comment has been added to explain the rationale of lack of curcumin nanocrystals in drug release and permeation studies: Although it is conceivable that curcumin is released more rapidly from the nanocrystals than when loaded into clay-based hydrogels, its application in form of nanocrystals is not feasible as they are in an aqueous solution consisting of 98.5% water, which would cause leakage from the application site resulting in poor drug absorption. At this purpose, clay minerals were used to overcome this drawback leading to an increase in the stability and viscosity of nanocrystals and improving curcumin bioavailability

4) For TGA and XRPD experiments: curcumin nanocrystals as well as the clays need to be run separately as well as a physical mixture ( simple powder-powder mixing)

  • TGA experiments of the pristine components have been added in the figure.

Figure 6. TGA curves of pristine components and curcumin nanocrystals-loaded hydrogels (B-CUR, P-CUR, and B/P-CUR).

  • As for XRPD analysis, the pristine components were previously studied in other works of ours. In particular, the spectra of CUR nanocrystals and its components including raw powder, Poloxamer 188, as well as the physical mixture have been reported by Casula et al (doi: 10.3390/pharmaceutics1
    Moreover, XRPD spectra of palygorskite (doi: org/10.1016/j.clay.2017.12.027) has been reported by Carazo et al, and bentonite (Doi: 10.2147/IJN.S208713.) by García-Villén et al. These works have been reported in the manuscript and a proper comment has been added

Round 2

Reviewer 1 Report

The manuscript has been properly revised. Although, still the presentation of results with uncertainties (Table 1) should be corrected. Significant digits are not the same as the number of decimal places. The following websites clarify this issue.

https://www2.southeastern.edu/Academics/Faculty/rallain/plab194/error.html

https://www.jyu.fi/science/en/physics/studies/student-laboratory/rules-for-presenting-the-measured-value-and-the-uncertainty

Author Response

English language and style are fine/minor spell check required.

The manuscript has been properly revised. Although, still the presentation of results with uncertainties (Table 1) should be corrected. Significant digits are not the same as the number of decimal places. The following websites clarify this issue. https://www2.southeastern.edu/Academics/Faculty/rallain/plab194/error.html, https://www.jyu.fi/science/en/physics/studies/student-laboratory/rules-for-presenting-the-measured-value-and-the-uncertainty

The authors wish to thank the reviewer for his/her comments to improve the paper quality. The significant digits have been corrected, and English language and style have been checked.

Reviewer 2 Report

The manuscript can be considered publishable

Author Response

English language and style are fine/minor spell check required. The manuscript can be considered publishable

The authors wish to thank the reviewer for his/her comments to improve the paper quality. English language and style have been checked.
